# Beyond the Reported Cutoff: Where Large Language Models Fall Short on Financial Knowledge

**Agam Shah**✉**, Liqin Ye**✉**, Sebastian Jaskowski, Wei Xu, & Sudheer Chava**
College of Computing & Scheller College of Business
Georgia Institute of Technology, Atlanta, GA
✉ Corresponding Authors: {ashah482, lye48}@gatech.edu

## Abstract

Large Language Models (LLMs) are frequently utilized as sources of knowledge for question-answering. While it is known that LLMs may lack access to real-time data or newer data produced after the model's cutoff date, it is less clear how their knowledge spans across *historical* information. In this study, we assess the breadth of LLMs' knowledge using financial data of U.S. publicly traded companies by evaluating more than 197k questions and comparing model responses to factual data. We further explore the impact of company characteristics, such as size, retail investment, institutional attention, and readability of financial filings, on the accuracy of knowledge represented in LLMs. Our results reveal that LLMs are less informed about past financial performance, but they display a stronger awareness of larger companies and more recent information. Interestingly, at the same time, our analysis also reveals that LLMs are more likely to hallucinate for larger companies, especially for data from more recent years. The code, prompts, and model outputs are available on GitHub.

## 1 Introduction

As research and development of Large Language Models (LLMs) continues to progress, evidence has emerged of heavy usage of these models within the financial domain. Cheng et al. (2024) find that trading volumes significantly decline during ChatGPT outages, indicating potential reliance on LLMs by the investors, and numerous research papers have been published suggesting that LLMs can provide investment advice or generate investment strategies (Romanko et al., 2023; Dong et al., 2024). Notably, Oehler & Horn (2024) commented that *"ChatGPT provides better financial advice for one-time investments than robo-advisors"* in regards to bond and equity index funds. Moreover, Yue et al. (2023) stated that LLMs can serve as a method to *democratize* financial knowledge. Such growing interest in LLMs in the financial industry warrants a closer investigation into the various biases and inaccuracies that may be present in these models.

In this work, we aim to identify and analyze these biases by examining where knowledge gaps in these models are present. While there has been considerable Natural Language Processing (NLP) research into gender biases (Lu et al., 2020), political biases (Motoki et al., 2023), and cultural biases (Naous et al., 2024), there has been limited study of LLM biases in the financial domain. We bridge this gap by considering how various factors such as time period, size, popularity among retail and institutional investors, market capitalization[1], and the readability of financial reports affect the answering capabilities of LLMs, and whether these knowledge gaps are biased along certain temporal or cross-sectional[2] factors. Our study evaluates leading language models, including `GPT-4o`, `GPT-4o-mini`, `GPT-4.5` (OpenAI, 2024), `Gemini` (Gemini Team et al., 2024), `DeepSeek-V3` (Liu et al., 2024), and `Llama-3-Chat` (Touvron et al., 2023), revealing temporal and cross-sectional knowledge gaps in LLMs.

---

[1]Definition and further reading for financial terminologies are provided in Appendix A.

[2]Cross-sectional analysis examines data from multiple companies at a single point in time.

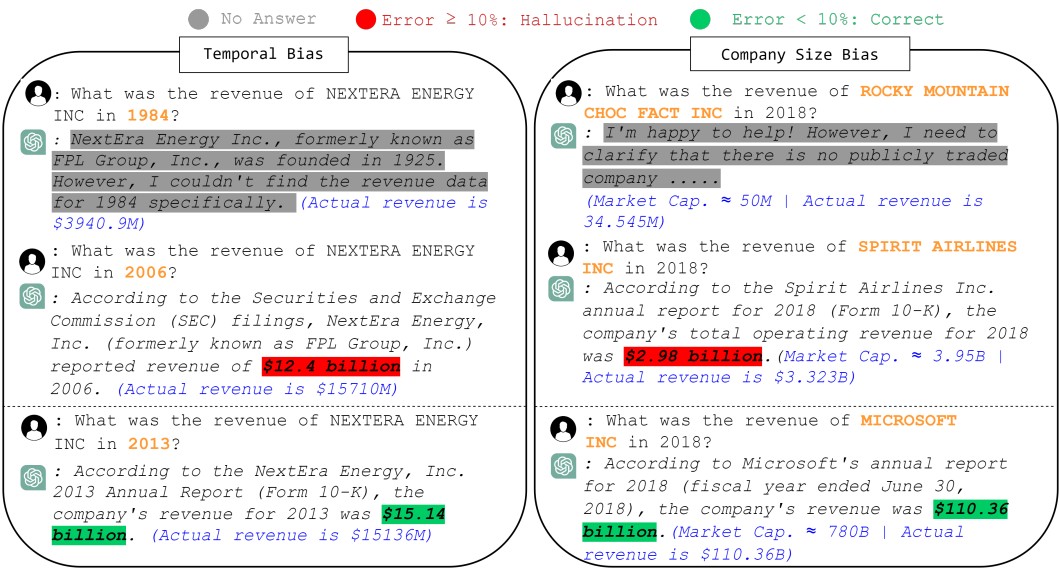

Figure 1: Example responses from `Llama3-70B` showcasing knowledge biases towards older (vs. newer) financial information and smaller (vs. bigger) companies.

Our analysis shows several key findings. We see that there is a possible *"retrograde"* knowledge bias, where LLMs are unable to answer financial questions *before* a certain time (e.g., what is a company's revenue back in the year 1984? See Figure 1). This complements existing research that highlights LLMs' struggles with knowledge produced *after* the cut-off dates of pre-training data (Onoe et al., 2022; Kasai et al., 2024). For instance, `Llama-3-70B-Chat` accurately answers revenue values for 54.17% of companies in 2017 while answering accurately for only 6.32% of companies in the year 1995 —despite financial data being publicly available for all U.S. companies since 1995.

We also find that LLMs demonstrate better accuracy for companies with larger market capitalizations, higher attention from both retail and institutional investors, higher number of SEC filing accesses, and better filing readability. For `Llama-3-70B-Chat`, a tenfold increase in market capitalizations of the company leads to a 1.0091 rise in the log odds ratio of accurately answering revenue-related questions. Such cross-sectional bias may lead to sub-optimal decision-making and unintended downstream effects in the markets. For example, a bias in LLMs towards favoring larger firms may ultimately cause reduced capital allocation to smaller firms (see Appendix K).

Additionally, we extend our analysis to examine how both time and firm factors relate to the model's propensity to produce *factuality hallucinations*, one of two types of hallucinations defined by Huang et al. (2023). Factuality hallucinations are defined as discrepancies between LLM-generated content and verifiable real-world facts. When the LLM response contains a numerical value, we can establish a numerical measurement to identify factuality hallucination. We calculate the error rates of LLM answers with respect to ground truth values and consider a response a hallucination if the error rate is above a certain threshold (10%). Interestingly, we find that hallucination occurs more in those firms which also have higher accuracy; for `Llama-3-70B-Chat`, a tenfold increase in market capitalizations results in a 0.1914 rise in the log odds ratio of hallucinating revenue—again, despite financial data being widely available.

As a result of this work, we put forth the following contributions:

- Utilizing diverse financial datasets to construct and analyze over 197k questions, providing insights into the performance and bias of leading LLMs.

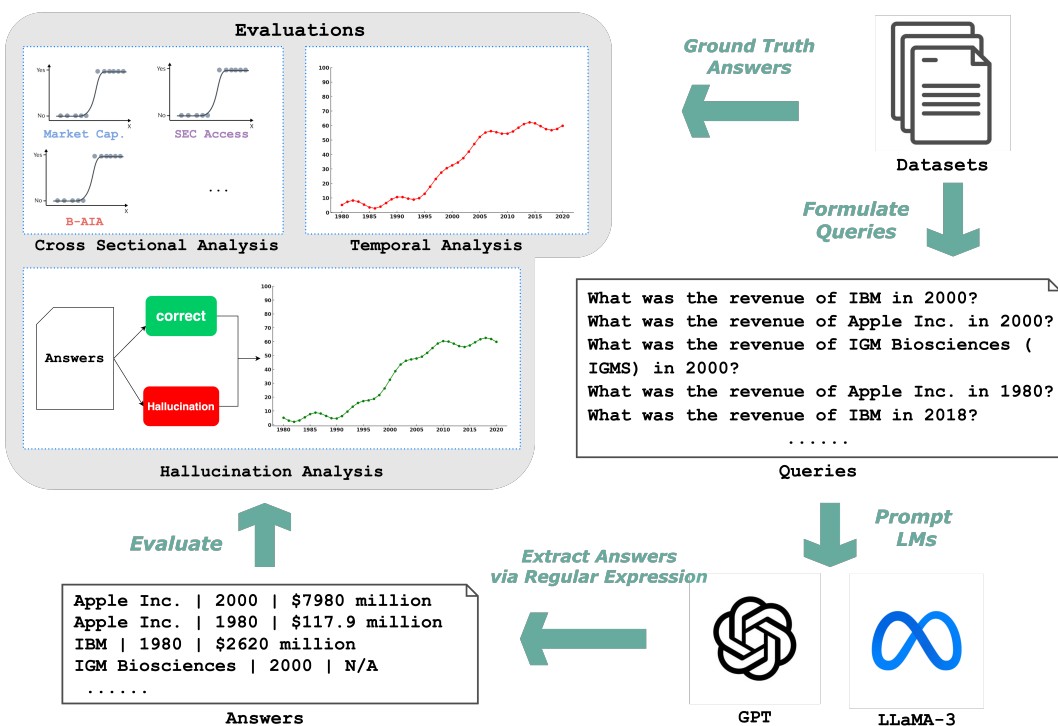

Figure 2: Experiment pipeline to understand and measure knowledge gap in LLMs.

- Introducing a novel, systematic method for analyzing temporal knowledge biases of LLMs. To the best of our knowledge, this is the first study to analyze the retrograde knowledge bias in LLMs.
- Utilizing our cross-sectional analysis framework to investigate a range of factors that are associated with knowledge bias in LLMs.
- Conducting a comprehensive hallucination analysis that identifies a notable trend of overconfidence in LLMs, particularly evident in responses concerning specific companies and time periods.

## 2 Experimental Dataset Construction

To facilitate our analysis, we construct a novel Revenue Prompt Dataset that consists of over 197k question-answer pairs about more than 17k unique companies, spanning over a 43-year period (Section 2.1). We pair this dataset with a diverse set of company characteristic variables to further study the effect of cross-sectional variables (Section 2.2).

### 2.1 Revenue Prompt Dataset (RPD)

We extract 197,011 revenue data points from the Compustat-IQ dataset (S&P Global Market Intelligence, 2023), spanning the years from 1980 to 2022. Each data point is a triple of the company name, year, and revenue (in million USD). Our dataset contains 17,621 unique companies listed on stock exchanges[3] in the United States, including those that have entered or exited the market due to IPOs, bankruptcy, or privatization. The change in the number of companies across years in our dataset is shown in Figure 3.

We use the entire dataset of 197k samples in most of our analyses for relatively less expensive models (`GPT-4o-2024-08-06`, `GPT-4o-mini-2024-07-18`, `Llama-3-8B-Chat`,

---

[3]The data covers 13 different stock exchanges including NYSE and Nasdaq

and `Llama-3-70B-Chat`), by asking LLMs to answer questions about the revenue of a company of a given year (Section 3). For the more costly (`GPT-4.5-preview-2025-02-27`, `DeepSeek-V3`, and `Gemini 1.5 Pro`) models, we use a subset ($RPD_{200perYear}$) of 200 firms per year from the full sample. In addition, we create another subset ($RPD_{430}$) with 430 companies that have a long public history with records available for every year between 1980 and 2022. These subsets of RPD are discussed in further detail in Appendix B.2.

## 2.2 Cross-Sectional Variables

Besides revenue, we also draw data from multiple financial resources to consider other key variables, as summarized in Table 1. These variables include market capitalization, attention from retail investors, attention from institutional investors, the frequency of regulatory filing downloads from SEC databases, and the readability of these SEC filings. We use standard company identifiers, like the SEC's Central Index Key (CIK), to align this data with our revenue data. By examining these diverse and multifaceted factors (Section 4), we aim to shed light on the various elements that potentially shape the knowledge bias in LLMs.

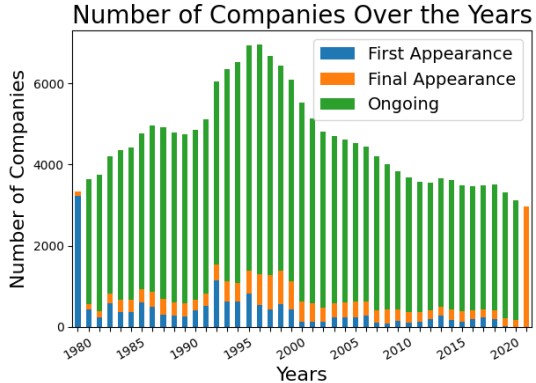

Figure 3: Number of companies from 1980 to 2022. 'First Appearance' counts companies entering the sample each year. 'Final Appearance' indicates companies exiting or appearing only once. 'Ongoing' tracks companies remaining in future years.

**Market Capitalization (MCap):** We collect data on the stock price and number of outstanding shares from the "Monthly Stock File" database of The Center for Research in Security Prices, LLC (CRSP), which is shared by Wharton Research Data Services. We cover the data starting in 1980 and ending in 2022. After collecting the data we calculate the market capitalization for a company for a given day by multiplying the number of outstanding shares with the price of the share on the day. For each financial year, we use the market cap value on the last trading day as the value for that year. The market cap values are then adjusted for inflation, and a logarithmic transformation is applied for better scaling (further details in B.1).

| Data (Source) | Key Variables | Years | #Data Points | #Unique Companies |
|---|---|---|---|---|
| Annual Financials (Compustat Capital-IQ) | Revenue | 1980-2022 | 197,011 | 17,621 |
| Market Capitalization (CRSP MSF) | Price and # Shares | 1980-2022 | 197,011 | 17,621 |
| Robinhood (Robintrack) | # Holders | 2018-200 | 9,592 | 3,469 |
| Bloomberg AIA (Bloomberg) | B-AIA | 2010-2020 | 21,303 | 2,432 |
| SEC Access (SEC-EDGAR) | # Access | 2003-2017 | 58,519 | 7,581 |
| Bog Index (Bonsall IV et al., 2017) | Bog Index | 1994-2020 | 110,161 | 12,822 |

Table 1: We utilize a wide range of finance datasets from notable sources. More details about each dataset can be found in the Appendix B.1.

**Robinhood Retail Attention:** To understand how the popularity of the stocks among retail investors correlates with the model's ability to answer the question for those companies, we collect from Robintrack, a dataset produced by the popular stock trading platform Robinhood. Robintrack keeps track of how many Robinhood users hold a particular stock over time. In particular, we collect the popularity metric which represents the number of unique user accounts that hold at least one share of the stock. To ensure comparability with the other variables in our analysis, we apply standardization to the data, involving mean subtraction followed by division by the standard deviation. We utilize each firm's Ticker (a

short string used to identify a firm listed on an exchange) for matching with other datasets. We have this data available only from 2018 to 2020.

**Bloomberg Abnormal Institutional Attention (B-AIA):** Our measure of institutional attention, sourced from Bloomberg, caters to a limited audience of around 320,000 subscribers, mainly institutional investors, who are subject to the high cost of $24,000 per annual subscription. We collect the data from Bloomberg Terminal which we have accessed through our institution. We utilize methodology used in Chava & Paradkar (2016), which assigns an hourly attention score for each stock by comparing the average count of news article views and searches over the past 8 hours to the distributions of these values over the past 30 days. By aggregating the highest hourly attention scores into a daily attention score, we gain insights into the institutional investor interest. We apply standardization to the data similar to Robinhood data. Refer to Appendix B.1 for more details.

**SEC Access:** SEC filings include financial statements and other formal documents submitted to the U.S Securities and Exchange Commission (SEC). As a proxy for a company's popularity among investors, we utilize over 58,000 searches on SEC.gov from February 2003 to June 2017, recorded in the EDGAR Log File, converting queries into CIK–year pairs to measure average daily filing accesses. We apply standardization to the data similar to Robinhood data.

**Bog Index for Readability:** We also consider the readability of the financial reports companies filed with the SEC to explore how variations in readability may correlate with language models' ability to answer questions about those companies. We utilize the Bog Index (Bonsall IV et al., 2017) readability score, which is commonly used for finance fillings (Bonsall et al., 2017), for the 110,161 SEC 10-K filings from 12,822 companies between 1994 and 2020 in the Bog Dataset. A higher score equates to a less readable document.

## 3 Temporal Finance Knowledge Gaps in LLMs

Since 1995, annual regulatory filings of publicly traded companies in the U.S. have been transparently available in the Securities and Exchange Commission (SEC)'s EDGAR database. Given this, we postulate that the models might have an ascending knowledge proficiency from 1995 onwards, as exemplified with NextEra Energy, Inc. in Figure 1 (left). We use our prompting dataset (RPD) to test four LLMs (`GPT-4o-2024-08-06`, `GPT-4o-mini-2024-07-18`, `Llama-3-8B-Chat`, and `Llama-3-70B-Chat`) on their temporal knowledge gap, checking if these models have greater revenue knowledge for more recent years, especially years after 1995.

**Revenue QA Prompts.** We construct LLM prompts (Figure 2) programmatically using a template: *What was the revenue of {company_name} in financial year {financial_year}?* This simplistic prompt design is motivated by the observation that non-professional investors with limited familiarity with AI technology are inclined to rely on readily accessible LLMs. Given the potential for inaccuracies in these models, such vulnerable populations could face significant adverse consequences (Chava et al., 2022). While more extensive prompt engineering with few-shot demonstrations may yield more precise answers, many general users may not employ sophisticated prompting strategies or engage in further fine-tuning of the models. We thus perform zero-shot prompting on the four LLMs with a *temperature* value of 0.00 (for reproducibility) and *max_token* value of 100. Based on the output of the model, we utilize regex to extract the numerical value of revenue, if it is available, and standardize currency units to millions of US dollars. Further implementation details for models are provided in Appendix C, while the effectiveness of regex is discussed in Appendix D. More discussion on why we choose "revenue" as a question is provided in the Appendix J. Additionally, more sophisticated revenue prompting using Chain-of-Thought for stock recommendation is detailed in Appendix K.

**Metrics for Temporal Analysis.** We use a ternary outcome variable $Y_{i,t}$ for firm $i$ and year $t$ based on the model's answer to evaluate the model performance. The variable $Y_{i,t}$ takes

the value of 2 if an absolute % difference of the numerical answer of revenue is less than 10% away from the ground truth as **a success**, 1 if an absolute % difference of the answer is greater than 10% as **a factual hallucination**, and 0 if no numeric answer is provided. In Appendix E, we show that our results are consistent even if this error threshold is varied.

$$Y_{i,t} = \begin{cases} 2: & \text{absolute \% error} < 10\% \\ 1: & \text{absolute \% error} \geq 10\% \\ 0: & \text{no numerical answer} \end{cases}$$

We analyze the temporal trend over the years by calculating the success rate and hallucination rate. For each given year, the success rate is a ratio of the number of successful responses to the total number of data points, while the hallucination rate is a ratio of the number of hallucinatory responses to the number of data points where the model fails. For clarity, we define both success rate and hallucination rate in Equation 1 below.

$$\text{Success Rate (T)} = \frac{\sum_{i,t=T} \mathbb{1}_{\{Y_{i,t}=2\}}}{\sum_{i,t=T} 1}$$

$$\text{Hallucination Rate (T)} = \frac{\sum_{i,t=T} \mathbb{1}_{\{Y_{i,t}=1\}}}{\sum_{i,t=T} \mathbb{1}_{\{Y_{i,t} \neq 2\}}} \tag{1}$$

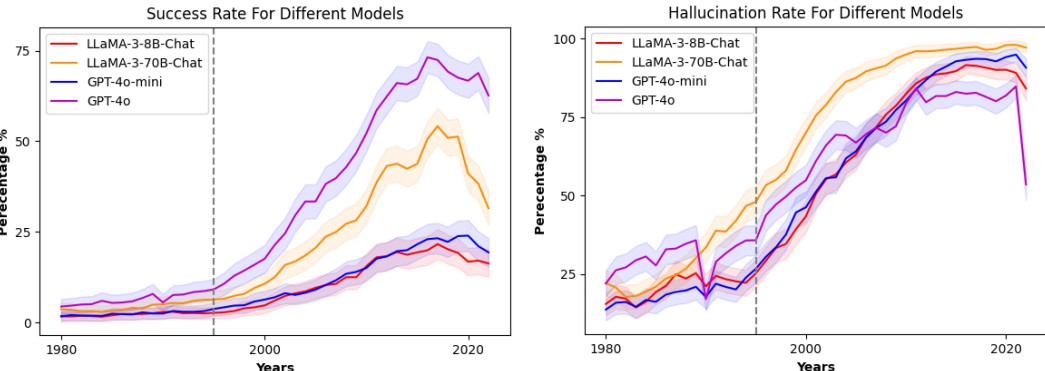

Figure 4: Success and hallucination rates of GPTs and Llamas over time. The dotted line is drawn at the year 1995. The shadow area around the line is the standard deviation of model performance.

**Results.** In Figure 4 (left), we present the temporal success rate trends of four models. This result underscores our claim that LLMs demonstrate a heightened proficiency in answering questions from more recent years as opposed to earlier ones. It's pertinent to highlight the dotted line at the year 1995, signifying the inception of the SEC's EDGAR filing system. After this date, detailed financial information from US public companies became publicly accessible online, thus augmenting the datasets available for model training. Intriguingly, there is a noticeable dip in the performance of all models in the years 2021 and 2022 when compared against their performance in 2018 to 2020. Similar results for GPT-4.5, DeepSeek-V3, and Gemini 1.5 Pro on RPD$_{200perYear}$ sample are presented in Figure 7 of Appendix F. Investigating the underlying reason behind this deviation promises to be a compelling avenue for further research.

In Figure 4 (right), we present the percentage of companies for which the model hallucinates revenue information over the years. Among the four models, Llama-3-70B-Chat hallucinates at a much higher rate over recent years. Interestingly, these models are more likely to hallucinate for the same years where they are also more likely to provide the correct answer.

## 4 Measuring LLMs' Cross-Sectional Biases in Finance

In addition to temporal knowledge gap biases (Section 3), we also explored how various cross-sectional factors affect the distribution of knowledge gaps. In financial literature, there have been studies showing a relation between analysts' biases and the size of the firm (Van Binsbergen et al., 2023) and the role of institutional attention (Ben-Rephael et al., 2017) in helping incorporate information in asset prices. Nevertheless, to the best of our knowledge, there is no study analyzing the relationship between firm-level information (size, institutional attention, popularity, etc.) and knowledge biases in LLMs. We extend our analysis to address this research gap and further explore how cross-sectional factors affect LLM knowledge and hallucination rates.

### 4.1 Metrics for Cross-Sectional Analysis

We use the same ternary outcome variable $Y_{i,t}$ that indicates model performance as discussed in Section 3 for our temporal analysis. To capture the relationship between the model's answer and company-level characteristics, we run the following logistical regression with $X_{i,t}$ as company-level characteristics:

$$\text{logit}(P(Y_{i,t} = y)) = \alpha + \beta * X_{i,t} + \delta_t * D_t + \epsilon_{i,t} \tag{2}$$

Here $Y_{i,t}$ is the outcome variable where $y = 2$ indicating model success while $y = 1$ indicating hallucination, $\delta_t$ is a year-fixed effect, $\alpha$ is a constant term, and $\epsilon_{i,t}$ is an error term. The coefficient $\beta$ will help us understand the influence of company-level characteristic $X_{i,t}$ on the outcome variable $(Y_{i,t})$.

### 4.2 Results and Analyses

| | Llama-3-8B | | Llama-3-70B | | GPT-4o-mini | | GPT-4o | |
|---|---|---|---|---|---|---|---|---|
| $X_{i,t}$ | $\alpha$ | $\beta$ | $\alpha$ | $\beta$ | $\alpha$ | $\beta$ | $\alpha$ | $\beta$ |
| MCap (log) | -13.0984‡ | 1.0546‡ | -11.8915‡ | 1.0091‡ | -15.1833‡ | 1.2879‡ | -10.9046‡ | 0.9209‡ |
| retail_inv (std) | -1.2811‡ | 0.2073‡ | 0.1755‡ | 0.1036‡ | -1.1033‡ | 0.5727‡ | 0.9562‡ | 0.2589‡ |
| B-AIA (std) | -1.1272‡ | 0.0187 | -0.1870‡ | 0.0107† | -1.0425‡ | 0.0217† | 0.6939‡ | 0.0180† |
| SEC-Access (std) | -2.4464‡ | 0.0628‡ | -1.5922‡ | 0.0782‡ | -2.4840‡ | 0.1028‡ | -0.8490‡ | 0.0770‡ |
| Bog Index (std) | -2.5228‡ | -0.0965‡ | -1.8361‡ | -0.0666‡ | -2.4133‡ | -0.0296‡ | -1.5021‡ | -0.0753‡ |

Table 2: Empirical cross-sectional regression results for market cap (MCap), number of retail investors on Robinhood (retail_inv), Bloomberg abnormal institutional attention (B-AIA), number of access on SEC-EDGAR (SEC-Access), and measure of readability (Bog Index). "std" in the parentheses indicates the data is standard normalized. *, †, and ‡ indicate p-value at the 10%, 5%, and 1% levels, respectively for regression coefficients.

To investigate the influence of a company's market capitalization (size) on the proficiency of LLMs in answering the company's financial details, we conducted a logistic regression analysis as outlined in Equation 2. In this regression, we adopted the logarithm (base 10) of the company's market cap as the independent variable, denoted as $X_{i,t}$. The outcomes of this regression analysis are presented in Table 2. The data suggests that LLMs exhibit enhanced performance for companies with larger market capitalization. A tenfold increase in the market cap corresponds to an increment of 1.2879 in the log odds ratio of the GPT-4o-mini model answering revenue-related questions

| Model | Constant ($\alpha$) | Beta ($\beta$) |
|---|---|---|
| Llama-3-8B | -6.7754‡ | 0.6053‡ |
| Llama-3-70B | -2.8859‡ | 0.1914‡ |
| GPT-4o-mini | -6.2188‡ | 0.5222‡ |
| GPT-4o | -2.1171‡ | 0.0964‡ |

Table 3: Market cap analysis results on hallucination based on the empirical regression. *, †, and ‡ indicate p-value at the 10%, 5%, and 1% levels, respectively for regression coefficients.

correctly, and an increase of 1.0546, 1.091 and 0.9209 for the `Llama-3-8B-Chat`, `Llama-3-70B-Chat`, and `GPT-4o` models respectively. Similar results for `GPT-4.5`, `DeepSeek-V3`, and `Gemini 1.5 Pro` on RPD$_{200perYear}$ sample are presented in Table 6 of Appendix G. The Results of Bloomberg AIA, SEC Access, and retail investment from Robinhood data follow a similar trend. As a higher value of the Bog Index represents that the filing is less readable, the variable has a negative beta indicating that the lower readability leads to a decrease in the performance of LLMs.

We similarly regress hallucination on the log market cap using Equation 2 by setting y=1. Based on the results in Table 3, we find that the model is more likely to hallucinate for the companies with higher market cap. Similar results for `GPT-4.5`, `DeepSeek-V3`, and `Gemini 1.5 Pro` on RPD$_{200perYear}$ sample are presented in Table 6 of Appendix G. Combining the result of Table 2 and Table 3, we can hypothesize that the model is more likely to hallucinate for the same companies for which it is also more likely to provide the correct answer (have knowledge).

For further analysis, we count the number of years for which a given company produces LLM hallucinations and the number of years for which it provides the correct answer. The visualization of such a relationship for `Llama-3-70B-Chat` in Figure 5, shows that the revenue of companies with a higher market cap (indicated with a color close to light green) have a higher propensity to be answered correctly and hallucinated at the same time. Similar results for `GPT-4o` are provided in Figure 8 of Appendix H for the robustness of the finding. We hypothesize these results might be due to the model having higher confidence for relatively recent years and those higher market cap companies.

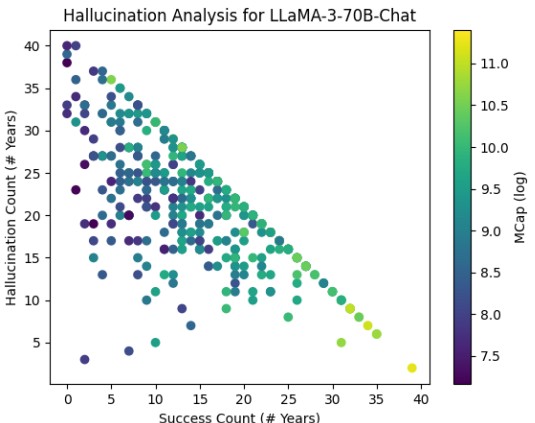

Figure 5: Each dot represents a company with a count on the x-axis indicating the number of years for which `Llama-3-70B` gave the correct answer and the y-axis indicating the number of years for which the model hallucinated in the answer. We take an average of market cap to assign a color to the dot.

## 5 Discussion

Our results highlight significant biases in the financial knowledge gaps of LLMs, along both temporal and cross-sectional dimensions. These biases are likely to have a substantial effect on the efficacy of LLM use both in the financial domain and across other domains as well. In addition, as LLMs continue to be touted as a *"democratizing"* force for financial knowledge, it is important to consider what effects biases may have on this process.

**Investors vs. Researchers:** Investors employing LLMs should recognize the bias risks highlighted in our work and avoid relying on LLMs for factual information or open-ended analysis. Biases identified, such as favoring larger firms, may unintentionally reduce capital allocation to smaller firms (see Appendix K). These biases pose particular harm to inexperienced investors, who may struggle to detect inaccuracies from LLM-generated content. For instance, Kumar (2009) finds younger investors prone to investing in "lottery-type" stocks, making them especially susceptible to biases in LLM-driven advice. Financial literacy educators should clearly communicate these bias-related risks in their teachings.

Researchers studying applications of LLMs in finance should carefully consider the various strata their tasks span, ensuring comprehensive experimental coverage. They should include firms across diverse years, sizes, and prominence. Attention to stratified biases strengthens results and ensures findings are not artifacts of model biases or limited to specific strata.

**Broader Impacts:**    While our work focuses on finance, our temporal and cross-sectional evaluation framework is widely applicable to assessing LLM knowledge across diverse domains. Many knowledge-intensive LLM tasks can be analyzed temporally or across specific entity-based strata. For example, in legal applications, one could evaluate LLM performance on laws enacted in different periods (e.g., comparing accuracy on laws from 2018 versus 1890) or laws receiving varying media coverage.

To illustrate our framework's versatility, we evaluate `Llama3-70B` on answering questions about annual La Liga soccer championship statistics. Specifically, we assess accuracy on questions like: *"In the years La Liga season, how many points did team finish with?"*. We observe higher accuracy for recent seasons and more prominent clubs, indicating biases toward recent data and larger entities. This demonstrates our evaluation framework's broader applicability; detailed methodology and results are presented in Appendix L.

## 6    Related Work

We briefly review existing research on societal and political biases, as well as hallucinations in LLMs. Our study uniquely contributes as the first, to our knowledge, to retrospectively assess knowledge biases in LLMs. Moreover, our research pioneers the exploration of various factors influencing financial knowledge bias in these models.

**Biases in LLMs:**    There has been much research involving the evaluation of LLMs' societal bias, such as gender, religion, race, politics, etc (Zhao et al., 2017; Lu et al., 2020; Sheng et al., 2021; Nozza et al., 2022; Kotek et al., 2023; Zhao et al., 2023; Blodgett et al., 2020; Galarnyk et al., 2025). Inspired by software testing, Nozza et al. (2022) proposed a systematic way of integrating societal bias testing into development pipelines. Zhao et al. (2023) unraveled the LLMs' "re-judgment inconsistency" in bias evaluation by leveraging psychological theories. Gender bias, as the issue most frequently addressed, has been resolved by diversified methods. Zhao et al. (2018) curated a new benchmark dataset WinoBias for testing and dispelling gender bias in LLMs. Kotek et al. (2023) use a simpler paradigm to identify gender bias and reveal that LLMs still tend to reflect the imbalance of their training dataset even after aligning with human preference. Additionally, Motoki et al. (2023) extends bias evaluations to political domains, revealing significant biases in ChatGPT towards certain political groups, necessitating further scrutiny measures in training processes.

**Hallucination in LLMs:**    Research on hallucinations in LLMs has produced several innovative methods for quantifying and addressing the issue. Zellers et al. (2019) introduced Grover, a model for generating and detecting "neural fake news," studying LLMs' tendency to hallucinate. Goyal & Durrett (2020) proposed assessing text-generation factuality by analyzing dependency-level entailment with source content. Lin et al. (2021) developed a dataset to test model truthfulness, indirectly evaluating hallucinations by alignment with facts. Huang et al. (2023) provides a comprehensive overview, categorizing hallucinations into factuality and faithfulness types. To our knowledge, our study is the first to measure hallucination in LLMs within the financial domain.

**LLMs in Finance**    Recent advancements in LLMs have significantly impacted the financial domain. Nie et al. (2024) introduced CFinBench, a comprehensive Chinese financial benchmark designed to assess LLMs' financial knowledge across various categories, including financial subjects, qualifications, practices, and laws. Similarly, Xie et al. (2023) developed PIXIU, a framework comprising a financial LLM fine-tuned with instruction data, alongside an evaluation benchmark covering various financial tasks. Kosireddy et al. (2024) explored the readiness of small language models for democratizing financial literacy, analyzing their performance in financial question-answering tasks. Additionally, Shah & Chava (2023) benchmarked the zero-shot performance of LLMs like ChatGPT on financial tasks, comparing them with fine-tuned models to assess their effectiveness in the financial domain. Collectively, these studies underscore the growing role of LLMs in enhancing financial applications and literacy.

## 7 Conclusion

In this study, we examine both the temporal knowledge bias and bias across firm-characteristic strata in LLMs using financial data from U.S. publicly traded companies. Our findings reveal that LLMs are more proficient with recent financial information, especially after the 1995 introduction of the SEC's EDGAR filing system. However, there's an unexplained dip in performance for 2021 and 2022. Secondly, LLMs demonstrate better accuracy for companies with larger market capitalizations, higher retail investment, higher institutional attention, higher number of SEC filing access, and higher readability. We also find that in the years and companies for which LLMs are more likely to provide a correct answer, they are also more likely to hallucinate. In essence, while LLMs offer valuable insights, their limitations in financial knowledge necessitate careful usage, especially in professional financial domains. Future work should explore the reasons behind these trends and enhance LLMs' performance breadth, as well as expand our methodology to other domains. This study further contributes to the discourse on how the availability of the pre-training corpus of LLMs for independent scientific scrutiny can facilitate scientific advancement.

## Contribution

AS conceived the research direction, acquired data, developed the Python scripts for analysis, and conducted the analysis. LY and SJ developed the Python scripts for analysis and constructed the visualizations. AS, LY, SJ, and WX helped write the manuscript. WX and SC supervised and guided the project.

## Acknowledgements

We did not receive any specific funding for this work. We thank Arnav Hiray, Michael Galarnyk, and Rohan Ganduri for their comments.

## Ethics Statement

Our work adheres to ethical considerations, although we acknowledge certain biases and limitations in our study.

- *Data Ethics*: The data used in our study, which is derived from publicly available sources, does not raise ethical concerns. All raw data is obtained for public companies that are obligated to disclose information under the guidance of the SEC and are subject to public scrutiny.

- *Language Model Ethics*: The language models employed (with proper citation) in our research are publicly available and fall under license categories that permit their use for our intended purposes. While most models (Llamas) employed are publicly available, it is important to note that prompt answers of `GPT-4o` will be made public under OpenAI's terms of use. The terms of use of OpenAI do not allow the use of prompt outputs for building competing models. Given the nature of our data, we believe this condition does not diminish the use of our work. We acknowledge the environmental impact LLMs have but we believe that the impact from just inference is limited.

- *Dataset Ethics*: We will not make any raw data used for the project public but we will make all the revenue prompts (RPD) and their answers public on our GitHub repository.

- *Limitations*: We do not run the analysis for `GPT-4.5`, `DeepSeek-V3`, and `Gemini 1.5 Pro` on the full sample due to high API cost. While we provide a small analysis for Llama-based financial LLMs in Appendix F, We are unable to run analysis on finance domain-specific pre-trained LLMs like BloombergGPT (Wu et al., 2023) as these models are not publicly available.

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

## A    Financial Lexicon: Definitions and Further Reading

Below we provide standard definitions of the finance-related lexicon used in the paper with the reference to further readings.

- **Outstanding Shares**: The total number of shares that are currently owned by all shareholders, including share blocks held by institutional investors and restricted shares owned by the company's officers and insiders. Further reading: https://www.investopedia.com/terms/o/outstandingshares.asp

- **Market Capitalization (MCap)**: The total market value of a company's outstanding shares of stock. It is calculated by multiplying the current market price of one share by the total number of outstanding shares. Further reading: https://www.investopedia.com/terms/m/marketcapitalization.asp

- **Consumer Price Index (CPI)**: A measure that examines the weighted average of prices of a basket of consumer goods and services, such as transportation, food, and medical care. It is calculated by taking price changes for each item in the predetermined basket of goods and averaging them. Further reading: https://www.investopedia.com/terms/c/consumerpriceindex.asp

- **Revenue**: The total amount of money generated by the sale of goods or services related to the company's primary operations. Further reading: https://www.investopedia.com/terms/r/revenue.asp

- **Robinhood**: A brokerage firm that offers commission-free trading of stocks and exchange-traded funds. Further reading: https://robinhood.com/us/en/about-us/

- **Ticker**: A unique series of letters assigned to a security for trading purposes. It is also known as a stock symbol. Further reading: https://www.investopedia.com/terms/s/stocksymbol.asp

- **IPO (Initial Public Offering)**: The process by which a private company can go public by the sale of its stocks to the general public. It could be a new, young company or an older company that decides to be listed on an exchange and hence goes public. Further reading: https://www.investopedia.com/terms/i/ipo.asp

- **Bankruptcy**: A legal proceeding involving a person or business that is unable to repay their outstanding debts. The process begins with a petition filed by the debtor or on behalf of creditors. Further reading: https://www.investopedia.com/terms/b/bankruptcy.asp

- **Privatization**: The transfer of a business, industry, or service from public to private ownership and control. Further reading: https://www.investopedia.com/terms/p/privatization.asp

## B    Supplementary Data Information

### B.1    Details of Data Sources

- **Compustat Capital-IQ**: We only keep companies that provide consolidated (i.e. combined accounts for parent and subsidiary) financial statements and which report data in "Standardized" format according to Compustat – Capital IQ. This will exclude some companies located outside the U.S. (but listed in the U.S.) as they are not required to report consolidated financial statements.

- **Inflation Adjustments and Scaling for Market Capitalization**: For better scaling when running analysis, we convert the market capitalization values to log values with a base of 10. We then normalize these values to adjust for inflation using the Consumer Price Index (CPI) data collected from FRED (Federal Reserve Economic Data). The inflation-adjusted market cap is converted to values corresponding to December 2022. The formula used to adjust the market cap values of company $i$ on date $t$ for inflation is as follows:

$$MCap_{i,t,r} = MCap_{i,t,t} * \frac{CPI(r)}{CPI(t)}$$

where $MCap_{i,t,r}$ represents the adjusted market capitalization of company $i$ on date $r$, $MCap_{i,t,t}$ denotes the market capitalization of company $i$ as measured on date $t$, while $CPI(r)$ and $CPI(t)$ are the CPI for dates $r$ and $t$ respectively.

- **Robintrack**: In this data when we share it only includes normal, long shares (not options).

- **B-AIA**: The methodology employed to quantify institutional investor attention involves a process, as outlined by Bloomberg. Each news article read is assigned a score of 1, while searches are weighted more heavily at a score of 10. These activities are aggregated on an hourly basis, and the attention score is derived by comparing these hourly counts to the previous month's average, with adjustments for deviation levels. Scores range from 0 to 4, reflecting varying degrees of attention based on the percentiles of activity compared to the prior month. This process effectively captures the intensity of investor focus on a stock, with daily scores determined by the peak hourly attention score. For additional methodological details, refer to Chava & Paradkar (2016).

We use the following identifiers from datasets to merge them for analysis:

- Compustat Capital-IQ: GVKEY, Company Name, Ticker, CIK[4]
- CRSP MSF: PERMNO, Ticker, Company Name
- Robintrack: Ticker
- B-AIA: Ticker
- SEC Access: CIK
- Bog Index: GVKEY

### B.2 Data Samples

The sample size for each sample is listed in Table 4.

**RPD$_{200perYear}$:** Given the high API cost for GPT-4.5, DeepSeek-V3, and Gemini 1.5 Pro, we create a subset representative sample of our data. To do so we categorize the log market cap of the companies into 4 categories (i.e., <8.00, 8.xx, 9.xx, >=10.00)[5]. After that, we randomly sample 50 companies from each year and market cap categories making it 200 samples per year. For 43 years, the total number of samples will be 8,600 (200*43). The result of GPT-4.5, DeepSeek-V3, and Gemini 1.5 Pro on RPD$_{200perYear}$ sample is presented in Appendix F and G.

**RPD$_{430}$:** As new companies can go public and existing public can get delisted, the set of companies changes every year. To analyze the same set of companies over time, we created a sample of companies that were public for every year from 1980 to 2022. We have 430 companies in our sample. For 43 years, the total number of samples will be 18,490 (430*43). The result of these four models on RPD$_{430}$ sample is presented in Appendix I for the robustness check.

## C Model Implementation Details

All the API calls are made between February 15, 2025, and February 28, 2025. We ran inference on "Llama-3-8B-Chat" locally using Transformer (Wolf et al., 2020) library on

---

[4]The Central Index Key (CIK) is a unique identifier used by the SEC to identify filing companies.
[5]Here 8.00 corresponds to $100 million in 2021 value.

| Sample Type | Sample Size |
|---|---|
| Full Sample (RPD) | 197,011 |
| $RPD_{200 per Year}$ | 8,600 |
| $RPD_{430}$ | 18,490 |

Table 4: Sample size for different samples used in our analysis.

NVIDIA RTX A40 GPU. For the `"Llama-3-70B-Chat"`, and `DeepSeek-V3` inference, we use API from together.ai. We are grateful to them for providing free credits and making it possible.

## D  Manual Verification

We manually check the correctness of our prompting and regular expression in extracting revenue information from LLMs' outputs. We randomly sampled 100 GPT's answers in 2010. There are 85 numerical answers (answers containing numerical revenue) and 15 no-answers (answers containing no revenue). Our regular expression returns *None* for all no-answers and correctly retrieves all numerical revenue. For all 100 samples, we observe that the LLM does not have difficulty understanding the question.

## E  Robustness Error Threshold

We repeat both temporal and cross-sectional analysis by varying the error threshold of 5%, 10%, and 20%. The results in Figure 6 and Table 5 show that our findings are consistent.

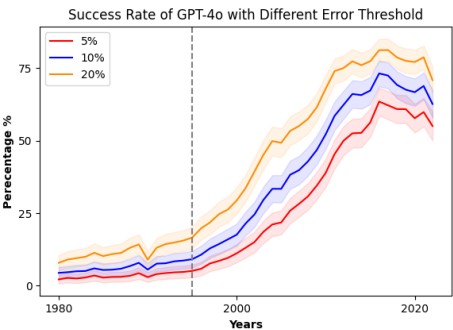

Figure 6: Performance of `GPT-4o` for three different error thresholds over time. The dotted line is drawn at the year 1995. The shadow area around the line is the standard deviation of model performance.

| Error Threshold | Constant ($\alpha$) | Beta ($\beta$) |
|---|---|---|
| 5% | -11.9616‡ | 0.9558‡ |
| 10% | -10.9046‡ | 0.9209‡ |
| 20% | -10.3530‡ | 0.9312‡ |

Table 5: Market cap analysis results on correctness based on the empirical regression for `GPT-4o` for three different threshold values of error. *, †, and ‡ indicate p-value at the 10%, 5%, and 1% levels, respectively for regression coefficients.

We calculate the raw error of `GPT-4o` over time and plot it in Figure 12. The color bar indicates the percentage of companies for which `GPT-4o` outputs an answer containing numerical revenue information for each specific year. The black dots outside the box are the outliers, representing `GPT-4o`'s hallucination. There are many more outliers in the year where `GPT-4o`'s answer rate is high, which is consistent with our findings in hallucination analysis: `GPT-4o` model is more likely to hallucinate for the year that it is also more likely to provide the correct answer. The overall trend of raw error also aligns with the error trend in our temporal analysis.

## F  Temporal Analysis of Additional Models on Small Sample

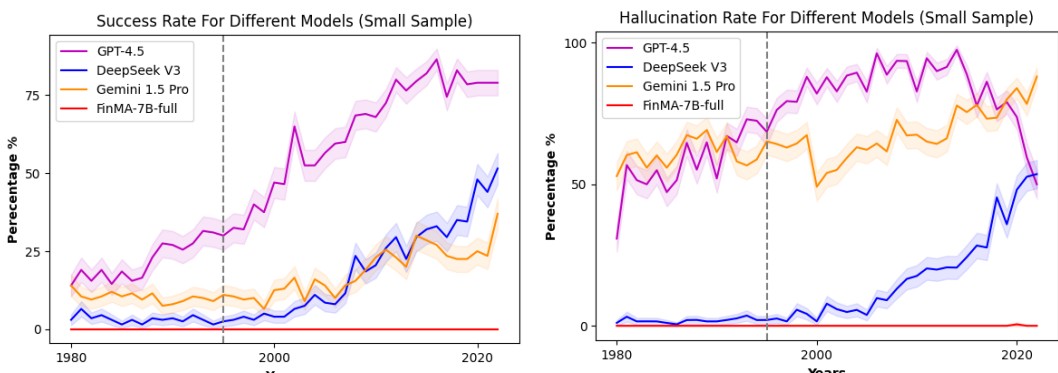

Figure 7: Success and hallucination rates of `GPT-4.5`, `Gemini 1.5 Pro`, `DeepSeek-V3`, and `FinMA-7B-full` over time. The dotted line is drawn at the year 1995. The shadow area around the line is the standard deviation of model performance. The performance is measured for all four models on RPD$_{200perYear}$ sample.

In Figure 7, we compare the performance of `GPT-4.5` ("gpt-4.5-preview-2025-02-27"), `Gemini 1.5 Pro` ("gemini-1.5-pro"), `DeepSeek-V3` ("DeepSeek-V3-FP8" on together-AI) and `FinMA-7B-full` ("TheFinAI/finma-7b-full"[6] from Xie et al. (2023)) over time. The performance is measured for all four models on RPD$_{200perYear}$ sample. The findings are similar to the results of Llamas and GPT-4o for the full sample reported in Figure 4 on the full sample. GPT-4.5 performs very similar to the GPT-4o not providing any significant gains for success, but performs better on hallucination for very recent years. We manually analyze the error of FinMA model, and find that almost all the time it provides either no text, or a number without unit. We exclude FinMA from any further analysis. For Gemini and DeepSeek, the trend is similar to the general trend.

## G  Cross-Sectional Analysis of Additional Models on Small Sample

The results for `GPT-4.5`, `Gemini 1.5 Pro`, and `DeepSeek-V3` on RPD$_{200perYear}$ sample in Table 6 are in accordance with the results of the main paper.

## H  Hallucination Additional Results

The results for `GPT4o` in Figure 8 are similar to the results for `Llama-3-70B-Chat` reported in Figure 5.

---

[6] https://huggingface.co/TheFinAI/finma-7b-full

| $X_{i,t}$ | DeepSeek V3 | | Gemini 1.5 Pro | | GPT-4.5 | |
|---|---|---|---|---|---|---|
| | $\alpha$ | $\beta$ | $\alpha$ | $\beta$ | $\alpha$ | $\beta$ |
| Success | -18.0719‡ | 1.5170‡ | -7.9137‡ | 0.6646‡ | -7.9533‡ | 0.6688‡ |
| Hallucination | -8.6866‡ | 0.4469‡ | -1.2108‡ | 0.1159‡ | -0.2099† | 0.0915‡ |

Table 6: Market cap analysis results on success and hallucination based on the empirical regression for GPT-4.5, Gemini 1.5 Pro, and DeepSeek-V3. *, †, and ‡ indicate p-value at the 10%, 5%, and 1% levels, respectively for regression coefficients. The study is conducted on RPD$_{200perYear}$ sample.

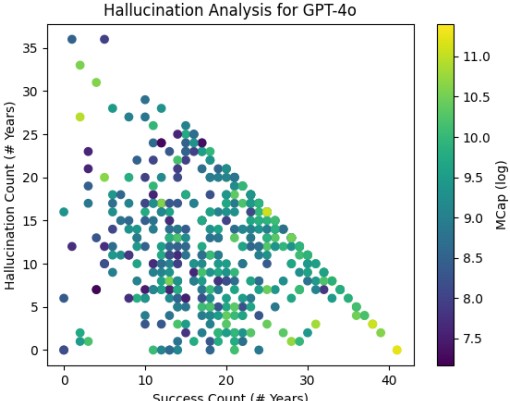

Figure 8: Each dot represents a company with a count on the x-axis indicating the number of years for which GPT-4o gave the correct answer and the y-axis indicating the number of years for which GPT-4o hallucinated in the answer. We take an average of market cap to assign a color to the dot.

## I  Robustness Check: Same Companies Over Time

Although our experiments provide substantial evidence showing LLMs' proficiency in answering financial questions for more recent periods and larger market cap companies, it is imperative to consider the potential confounding factors. The bankruptcy and establishment of companies throughout the years could introduce variability, thus potentially affecting the outcomes. Figure 9 indicates LLMs' temporal performance on 430 companies that have existed consistently over the 43 years in our sample, aligning consistently with our previous comprehensive companies analysis. Eliminating the bias from bankruptcy and IPOs, we can assert that LLMs exhibit enhanced capability in recent periods. The reason for enhanced performance compared to the full sample can be attributed to survival bias(Elton et al., 1996; Rohleder et al., 2011).

| Model | Constant ($\alpha$) | Beta ($\beta$) |
|---|---|---|
| Llama-3-8B | -10.4140‡ | 0.8408‡ |
| Llama-3-70B | -9.0624‡ | 0.7786‡ |
| GPT-4o-mini | -11.0263‡ | 0.9567‡ |
| GPT-4o | -8.4113‡ | 0.7051‡ |

Table 7: Market cap analysis results on RPD$_{430}$ over time based on the empirical regression. *, †, and ‡ indicate p-value at the 10%, 5%, and 1% levels, respectively for regression coefficients. The results are based on the full sample with year fixed effect.

In terms of market capitalization, we ran the same regression analysis on those companies. The result, displayed in Table 7, is similar to the previous market cap analysis. It reaffirms

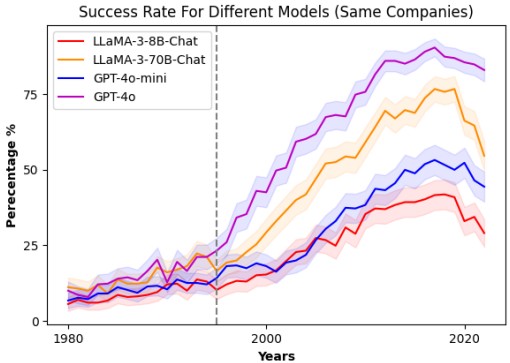

Figure 9: Performance of GPTs and Llamas on RPD$_{430}$. The dotted line is drawn at the year 1995. The shadow area around the line is the standard deviation of model performance.

the claim that the larger the company's market cap is, the more accurate LLMs' answers to its financial questions are.

## J  Why "Revenue"?

Below we provide further reasoning on why we picked "revenue" to form our question prompts.

- *Goal of our study*: As the goal of our study is to understand the temporal and cross-sectional biases, we want to ask questions that can vary across time as well as firms. Given our goal, it is not ideal to ask questions like "Can you explain what derivative securities mean?". The answer to this question is the same across time and firms.

- *Why not some financial ratio?*: Revenue is a top-line number that is directly available in SEC filings. If we form a question on a financial ratio like *return on assets*, even though we can validate the answer, it will involve two skills of model 1. ability to recall knowledge and 2. ability to calculate the ratio (do the arithmetic). As here we are focused on understanding the knowledge gap, a simple question on revenue is an appropriate choice.

- *Revenue Knowledge Bias Affects Stock Recommendation*: We conduct a study with Chain-of-Thought prompting to show that because of lack of knowledge of revenue for small companies, LLM (GPT-4o) is less likely to provide stock recommendation for them in Appendix K.

## K  News Prediction Analysis

To understand how the knowledge bias can relate to the model's inability to assess the impact of revenue forecast on the stock recommendation, we construct a Chain-of-Thought step-by-step prompt as shown in Figure 10 and ask GPT-4o ("gpt-4o-2024-08-06") to provide stock recommendation on whether to "BUY", "SELL" or "DNK" (do not have enough knowledge of the company). We filtered out the dataset according to the knowledge cut-off time of GPT-4o[7]. We also require that those companies have revenue information available to us from 2018 to 2023. In total we end up with 2,751 companies in our dataset for this study.

We then evaluate whether the model's recommendation was BUY or SELL if the output label is not DNK. We run the same empirical regression used earlier in the paper. In this case, the Y variable is assigned a value based on whether the recommendation is BUY, SELL, or DNK.

---

[7]We ask the revenue forecast and stock recommendation after Sep 30, 2023.

---

**Multi-turn Stock Recommendation Prompt**

**[SYSTEM INPUT]**
Forget all your previous instructions.

**[USER INPUT]**
What are the revenues of {company} for each finance year from {start_year} to {end_year}? Please return the revenue only.
**Example:** What are the revenues of APPLE INC. for each finance year from 2018 to 2022? Please return the revenue only.

**[EXPECTED OUTPUT]**
ONLY the revenue information for {company} for finance year from {start_year} to {end_year}.
**Example:** Here are the revenues of Apple Inc. for each financial year from 2018 to 2022: 2018: $265.6 billion; 2019: $260.2 billion; 2020: $274.5 billion; 2021: $365.8 billion; 2022: $394.3 billion

**[USER INPUT]**
Based on the revenue information above, please predict the revenue of {company} in finance year {end_year+1}. Please return the revenue only.
**Example:** Based on the revenue information above, please predict the revenue of APPLE INC. in finance year 2023. Please return the revenue only.

**[EXPECTED OUTPUT]**
ONLY the revenue information for {company} for finance year {end_year+1}.
**Example:** $423.1 billion

**[SYSTEM INPUT]**
Act as a financial expert with experience in stock recommendations.

**[USER INPUT]**
Based on the information above, give either BUY, SELL, or DNK (do not have enough knowledge of the company) recommendation for {company} in finance year {end_year+2}.
**Example:** Based on the information above, give either BUY, SELL, or DNK (do not have enough knowledge of the company) recommendation for APPLE INC. in finance year 2024.

**[EXPECTED OUTPUT]**
BUY, or SELL, or DNK
**Example:** BUY

---

Figure 10: Example of multi-turn Chain-of-Thoughts (CoT) prompts for stock recommendation under the impact of model's knowledge in revenue information.

| Model | Constant ($\alpha$) | Beta ($\beta$) |
|---|---|---|
| No Recommendation (DNK) | 1.4804‡ | -0.0654‡ |
| BUY Recommendation | -0.5766‡ | 0.0751‡ |
| SELL Recommendation | 0.1006‡ | -0.0103‡ |

Table 8: Market cap analysis results on a stock recommendation based on the empirical regression for GPT-4o. *, †, and ‡ indicate p-value at the 10%, 5%, and 1% levels, respectively for regression coefficients. The study is conducted on a list of companies we have in the years 2018-2023.

The result in Table 8 indicates that the model is less likely to make a decision for smaller market-cap companies while more likely to provide the label "BUY" for large market-cap companies. The result serves as a tiny experiment to show how the knowledge bias can impact biases in investment recommendations. We note that this is just a study on only one year of data due to reported cut-off.

## L    Spanish Soccer Leagues Statistics Study

In order to demonstrate the applicability of our LLM bias evaluation framework to non-finance domains, we examined the application of our framework to assess LLMs' abilities to answer questions about soccer statistics. We compiled a dataset of 1,676 samples containing season statistics for a given season and team in both the La Liga and Segunda División Spanish soccer leagues. Our dataset spans 41 seasons from 1980 to 2020 and contains statistics about 167 unique teams, though not every team is present every year, as teams are relegated down to or promoted from lower leagues based on their performance.

We then assess the ability of Llama3-70B (*temperature*=0, *max_tokens*=100) to answer the question: "*In the {year} {La Liga/Segunda División} season, how many points did {team} finish with?*". We assess the quality of responses across both the temporal and cross-sectional dimensions, looking at differences in model performance in different years, as well as differences in model performance for clubs with greater prominence (measured by finishing position). Model performance is measured using a success rate metric, which for this study is defined as the percentage of model responses which give a point value within 5% of the true point value.

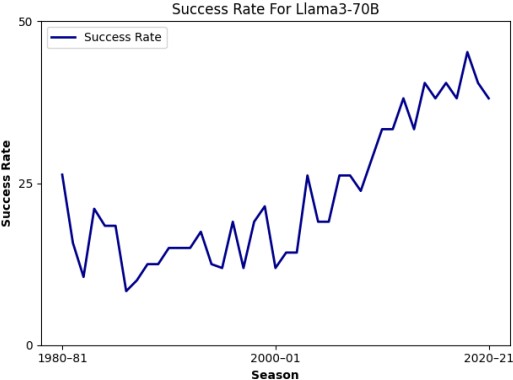

Figure 11: Success Rate of Point Answering Over Time.

Looking at the performance of the model, temporally (see Figure 11), we can see that the model performs better in more recent years, suggesting a bias towards more recent years.

Furthermore, we assess how the prominence of a soccer club affects the model's performance by running a logistic regression on $Pos_{i,t}$, the finishing position of each soccer club. Our logistic regression is defined in Equation 3.

$$\text{logit}(P(Y_{i,t} = y)) = \alpha + \beta * Pos_{i,t} + \delta_t * D_t + \gamma * \mathbb{1}_{\{league=Segunda\}} + \epsilon_{i,t} \tag{3}$$

Here $Y_{i,t}$ is the outcome variable where $y = 2$ indicating model success, $\delta_t$ is a year-fixed effect, $\alpha$ is a constant term, $\gamma$ is a league-fixed effect, and $\epsilon_{i,t}$ is an error term. We find $\gamma$ to be $-3.1317$, reflecting the fact that being in Segunda División decreases the probability of getting the correct answer compared to La Liga.

We find $\alpha = -1.2941$ and $\beta = -0.0542$ [8], suggesting that clubs with a greater finishing position (lower-prominent clubs) see a lower success rate in our experiment. This highlights a knowledge bias stratified along team prominence, where the LLM is more likely to have knowledge of statistics regarding more prominent and successful teams.

---

[8]Significance at the 1% level

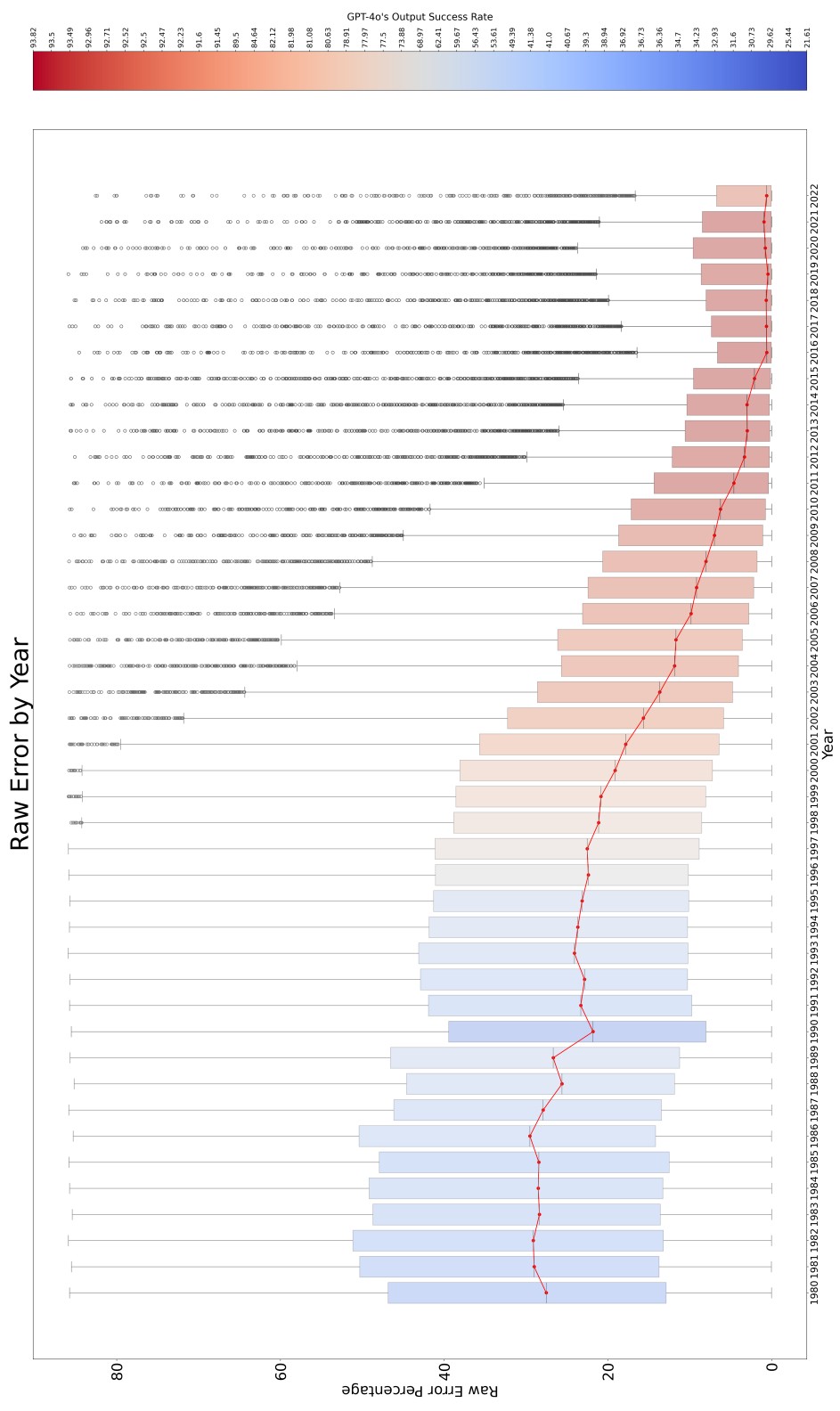

Figure 12: Raw answer error of GPT-4o ("gpt-4o-2024-08-06") over time.

