# OpenReview forum: "Beyond the Reported Cutoff: Where Large Language Models Fall Short on Financial Knowledge"
_colmweb.org/COLM/2025/Conference — COLM 2025_

### Official Review · Reviewer_xcLv · 2025-05-11

**Rating:** 4
**Confidence:** 4
**Ethics Flag:** 1

**Summary:**

In this paper, the authors investigate factors influencing the financial knowledge recall of large language models (LLMs). They find that LLMs exhibit more accurate financial knowledge regarding recent events, with a noticeable "cut-off" period before which factual recall significantly deteriorates. Additionally, the authors show that LLMs' factual recall tends to be biased toward companies with larger market capitalization. The paper clearly outlines the analytical methods and dataset construction procedures employed in their study.

**Reasons To Accept:**

1. The paper is clearly written and includes sufficient methodological details to allow other researchers to replicate or build upon the authors' work.

2. The in-depth analysis of this focused topic may be particularly valuable and of interest to researchers working in financial applications of large language models

**Reasons To Reject:**

1. The paper's contribution is incremental in nature. It lacks a novel approach or an innovative analytical method, and does not provide a new empirical application of natural language processing techniques to finance.

2. The method employed by the authors to verify the factual recall ability of the LLM involves prompting the model to recall company revenues for specific years. Although raw LLMs are frequently treated as knowledge bases in various scenarios, empirically, precise questions of this granularity typically require integration with external tools, such as retrieval-augmented generation or search engines, to achieve satisfactory accuracy. Analyzing raw model behavior alone may therefore offer limited insights into assessing or enhancing real-world system performance. Furthermore, focusing solely on revenue recall is only a narrow reflection of an LLM's overall financial knowledge, making it insufficient for drawing conclusive insights about the model’s broader financial knowledge capabilities.

3. The conclusions presented by the paper may lack depth. Given that an LLM’s knowledge originates from training data, it is expected that the model will exhibit a knowledge cut-off based on the availability of machine-readable data. Additionally, it is unsurprising that companies with larger market capitalization, receiving more extensive media coverage, would naturally have better representation in the LLM's knowledge base.

4. The target audience for this topic appears highly specialized within finance. While the analysis could certainly interest researchers in financial applications, it is unclear whether the topic aligns sufficiently with the broader interests of the general audience at COLM.

---

> ### Author Response · Authors · 2025-05-31
>
> We thank the reviewer for engaging with our paper and recognizing its clarity and methodological transparency. While we appreciate the feedback, we respectfully would like to clarify several aspects of the assessment. Our work intentionally focuses on a well-scoped and underexplored problem—temporal and cross-sectional financial knowledge gaps in LLMs—and makes methodological contributions that we believe are of independent value. Below, we clarify our motivations, address points of misunderstanding, and highlight how our results offer actionable insights for both researchers and practitioners in the LLM-for-finance community.
>
>
>
>  > “The paper's contribution is incremental in nature. It lacks a novel approach or an innovative analytical method, and does not provide a new empirical application of natural language processing techniques to finance.”
>
> **Response:** We respectfully disagree with the characterization of our contributions as incremental. To our knowledge, this is the first study to systematically assess **temporal and cross-sectional knowledge gaps in LLMs within the financial domain** using a large-scale, multi-decade QA benchmark. Prior work has primarily focused on post-cutoff date issues or benchmark performance on generic financial tasks, whereas our framework evaluates LLMs retrospectively—across time and firm characteristics—introducing a new lens for auditing factual knowledge in pretrained models.
>
> Our methodology includes:
>
>  - A 197K QA benchmark spanning 43 years and 17K+ companies;
>
>  - A cross-sectional regression framework analyzing biases across market cap, investor attention, and readability—factors previously unexplored in this context;
>
>  - A joint analysis of accuracy and hallucination behavior, revealing paradoxical overconfidence patterns in high-cap firms.
>
> Rather than presenting a new application of NLP to finance, our contribution lies in auditing the **epistemic boundaries of LLMs**—an increasingly vital area of inquiry as these models are adopted in high-stakes settings like finance. We believe this diagnostic perspective is both novel and broadly relevant.
>
>
>
>  > “Analyzing raw model behavior alone may offer limited insights... revenue recall is a narrow reflection of financial knowledge.”
>
> **Response:** We agree that real-world systems often benefit from retrieval components, but our goal is to **isolate the pretrained model's knowledge**—a standard and necessary baseline in LLM evaluation. Many high-stakes applications (e.g., retail investing tools) still rely on raw LLM outputs without augmentation, making this analysis highly relevant. We chose revenue precisely because it is a canonical, time- and firm-sensitive variable with ground truth available, enabling rigorous, large-scale evaluation.
>
>
>
>  > “The conclusions may lack depth… It is expected that models will have a knowledge cut-off and favor larger firms with more coverage.”
>
> **Response:** We would like to clarify that our contribution lies in **quantifying and verifying these assumptions systematically and at scale**, which had not been done before. Crucially, we show that **hallucination rates also rise for large-cap firms and recent years—contrary to expectations**—suggesting overconfidence where LLMs appear most reliable. This tension between knowledge recall and hallucination has direct implications for downstream use as demonstrated in Appendix K. We believe surfacing and formalizing this nuanced behavior adds both depth and value to the community’s understanding.
>
>
>
>  > “The target audience appears highly specialized within finance… unclear if the topic aligns with the broader COLM audience.”
>
> **Response:** While our primary evaluation is in finance, we explicitly demonstrate the **generality of our framework** through a domain-transfer study on **La Liga soccer statistics** (Discussion, Section 5; Appendix L). There, we find similar temporal and prominence-based knowledge biases in model performance—supporting the broader applicability of our approach. Our methodology is domain-agnostic and can be used to assess LLM factual knowledge across any structured entity-time setting (e.g., law, medicine, history), which we believe aligns well with COLM.
>
>
>
> We would also like to respectfully note that the **motivations, novelty, and broader applicability** of our framework are clearly outlined in the **Abstract, Introduction and Discussion** sections. We hope these clarifications help with a renewed assessment of the full scope and relevance of our contributions.

---

> > ### Comment · Reviewer_xcLv · 2025-06-02
> > **Response to Author Rebuttal**
> >
> > Thank you to the authors for your detailed and thoughtful responses. Please find my feedback on each point below:
> >
> > ## 1: Contribution and Novel Evaluation Perspective
> >
> > I some what agree that the contribution of the work on brining a new angle of evaluating the LLMs performance or introducing new resources given the dataset mentioned in the paper will be released.
> >
> > ## 2: Reliance on Parametric Knowledge vs. External Tools
> >
> > I would like to maintain my position on this point for further discussion. Regarding fine-grained knowledge such as specific revenue figures, I believe it is impractical to rely solely on the parametric knowledge of LLMs when a wide array of established tools (e.g., SQL databases, RAG systems) can retrieve such data with greater precision and reliability.
> >
> > Additionally, could the authors please provide references to support the claim that retail investing tools predominantly rely on the internal knowledge of LLMs?
> >
> > ## 3: Relationship Between Knowledge and Hallucination
> >
> > I would like to respectfully maintain my evaluation on this point, with two specific concerns:
> >
> > ### (a) Consistency with Paper Claims
> > I believe my original statement does not contradict the paper's hypothesis that "*we can hypothesize that the model is more likely to hallucinate for the same companies for which it is also more likely to provide the correct answer (have knowledge).*" and "The visualization of such a relationship for Llama-3-70B-Chat in Figure 5 shows that the revenue of companies with a higher market cap (indicated with a color close to light green) have a higher propensity to be answered correctly and hallucinated at the same time." The larger model tend to have better representation/knowledge for large cap companies, and it dose not necessary suggest that a richer data leads to hallucination.
> >
> > ### (b) Scope of Knowledge Deficiency Analysis
> > Given that the primary purpose of evaluating hallucination is to identify cases of LLM knowledge deficiency, could the authors elaborate on why the regression analysis only considers cases where the error exceeds 10% (y=1)? Wouldn't the absence of numerical responses also constitute a form of knowledge deficiency that merits analysis?
> >
> > I acknowledge that the authors' claims are well-founded and appreciate their responses. However, I would like to maintain my current evaluation pending further clarification on the points raised above.
> >
> > I look forward to the authors' additional responses and thank them again for their engagement with the review process.

---

> > ### Author Response · Authors · 2025-06-04
> >
> > We sincerely thank the reviewer for their thoughtful follow-up and for acknowledging the contribution of our dataset and evaluation perspective. We appreciate the opportunity to continue clarifying key aspects of our work. Below, we provide point-by-point responses to the comments.
> >
> > ## 1. Contribution and Novel Evaluation Perspective:
> >
> > We are grateful that the reviewer acknowledges the contribution of our evaluation framework and dataset release. We would like to emphasize that our core novelty lies in (a) framing *temporal and cross-sectional financial knowledge gaps* as an LLM evaluation problem and (b) showing how these gaps relate to both accuracy and hallucination behavior quantitatively and at scale.
> >
> > While grounded in finance, our approach is designed to be domain-agnostic and applicable to any setting with structured entity-time data. This generality is demonstrated in Appendix L through our soccer experiment, and we hope this further highlights the broader relevance of the proposed framework.
> >
> > ## 2. Reliance on Parametric Knowledge vs. External Tools:
> >
> > We appreciate the reviewer’s perspective and would like to clarify our intent. Our study does not argue that LLMs should replace retrieval-based tools for precise fact queries. Rather, our goal is to **probe and characterize the structure of parametric knowledge** in pretrained LLMs, specifically, how it varies over time and across firm characteristics. Furthermore, studies like Cheng et al. (2024) [1] also demonstrate that **LLMs do not always align with their stated knowledge cutoffs**. Our work complements these findings by offering a large-scale, controlled analysis of retrograde and cross-sectional knowledge gaps, a perspective that, to our knowledge, has not been previously formalized or quantified at this scale.
> >
> > Additionally, recent evidence suggests that even temporary disruptions in access to models like ChatGPT lead to significant drops in trading volume [2], indicating that investors do rely on raw LLM outputs in practice. This finding is based on data from November 2022 to December 2023, a period during which OpenAI’s web-browsing feature was disabled in July 2023 and reintroduced in late September 2023 only for paid users.
> >
> > While our core method focuses on revenue to enable clean evaluation of factual knowledge, we emphasize in **Section 5 and Appendix L** that our **evaluation framework is generalizable**, as illustrated through its application to Spanish soccer league statistics.
> >
> > [1] Cheng, Jeffrey, et al. "Dated data: Tracing knowledge cutoffs in large language models." arXiv preprint arXiv:2403.12958.
> >
> > [2] Qiang Cheng, Pengkai Lin, and Yue Zhao. Does generative ai facilitate investor trading? evidence from chatgpt outages. Evidence from ChatGPT Outages (June 21, 2024), 2024.
> >
> > ## 3. Relationship Between Knowledge and Hallucination:
> >
> > **(a) Consistency with Paper Claims**
> >
> > We would like to respectfully clarify that our earlier response was not intended to contradict the reviewer’s original statement. We fully agree that LLMs tend to have better knowledge of large-cap companies, a trend also supported by our results. Our intent was to respond to the earlier suggestion that “the conclusions may lack depth,” by highlighting that **our paper goes further than confirming expected trends.**
> >
> > Specifically, our analysis reveals a **counterintuitive relationship**: the same companies for which models exhibit stronger knowledge (e.g., high market cap firms) also see **higher hallucination rates**. This duality captured in Figure 5 and Table 3 was not obvious a priori and, to our knowledge, has not been empirically demonstrated at this scale. We believe this nuance adds depth and actionable insight.
> >
> > **(b) Scope of Knowledge Deficiency Analysis**
> >
> > We would like to clarify that our regression analysis in **Table 2** does account for both hallucination and no-answer cases as forms of failure. Specifically, as defined in the paper, a “success” is when the absolute error is below 10%, and all other outcomes, including the absence of a numerical answer, are treated as failures. Thus, our model performance analysis already covers the deficiency modes the reviewer refers to.
> >
> > Additionally, we provide a **robustness check in Appendix E**, showing that our key findings hold under varying error thresholds.
> >
> >
> >
> > We thank the reviewer again for their continued engagement and thoughtful questions. We hope these clarifications help illustrate the intent, depth, and generalizability of our work.

---

> > > ### Comment · Reviewer_xcLv · 2025-06-09
> > >
> > > Thank authors for the detailed responses, I find myself somewhat convinced by their approach and conclusions. I have accordingly revised my scores upward to better reflect this assessment.
> > >
> > > I think it would be beneficial if the authors could change the line 224 - 225 to reflect more explicitly reflect the cases included.

---

> > > > ### Author Response · Authors · 2025-06-09
> > > >
> > > > We sincerely thank the reviewer for revisiting our paper and for the constructive engagement throughout the discussion period. We are glad to hear that our clarifications were helpful and appreciate the updated assessment.
> > > >
> > > > We will revise lines 224–225 in the final version to more explicitly (using equations, as shown below in LaTeX) reflect the cases included in the regression, as suggested.
> > > >
> > > > \begin{equation}
> > > >     \begin{aligned}
> > > >     \text{Success Rate (T)} = \frac{\sum_{i, t=T}\mathbbm{1}_{\{Y_{i,t}=2\}}}{\sum_{i, t=T} 1} \\\\
> > > >     \text{Hallucination Rate (T)} = \frac{\sum_{i, t=T}\mathbbm{1}_{\{Y_{i,t}=1\}}}{\sum_{i, t=T}\mathbbm{1}_{\{Y_{i,t}
> > > > \neq2\}}}
> > > >     \end{aligned}
> > > > \end{equation}

---

### Official Review · Reviewer_2yqz · 2025-05-12

**Rating:** 6
**Confidence:** 5
**Ethics Flag:** 1

**Summary:**

This paper investigates temporal and cross-sectional knowledge biases of large language models (LLMs) in the financial domain. Authors construct a test set of 197K company-year revenue queries to evaluate several LLMs such as GPT-4o, Llama , DeepSeek-V3. The results show that LLMs perform  better on more recent years and on larger, more prominent companies, while hallucinations also increase for such companies. The paper also studies cross-sectional factors like market cap and show they they can predict both success and hallucination rate.

**Reasons To Accept:**

- Construction of a large-scale dataset (197K QA pairs) enabling  analysis of LLM financial knowledge over four decades.
- A systematic study of “retrograde” knowledge gaps and cross-sectional biases in finance.
- Providing evidence that LLMs favor recent data and large companies.
- Providing proof that the same reasons that drive success rate, also increases hallucination

**Reasons To Reject:**

The study still remains very shallow to be published in an LLM conference, and there is a lack of connecting the results to the theory of LLM and possibly to the architectural choices of LLMs studied.

Studied LLMs are still generalist one, while there exists many other more finance-specialized one which should have been taken into account in the study.

---

> ### Author Response · Authors · 2025-05-31
>
> We thank the reviewer for their thoughtful assessment of our work and for recognizing the contributions we make, including the construction of a large-scale dataset, the study of retrograde and cross-sectional knowledge biases, and the insights into how the same factors that drive model accuracy also increase hallucination rates. We are encouraged by the acknowledgment that these findings are important and novel, and we appreciate the opportunity to address the constructive critiques raised.
>
>
>
> **Theory of LLM**:
>
> We respectfully clarify that our work is not intended as a theoretical or architectural study of LLMs. Rather, our goal is to empirically uncover structured biases in model behavior—specifically, how temporal and cross-sectional financial factors influence both model success and hallucination. As noted in the Discussion section (section 5), we believe establishing these empirical regularities is a critical prerequisite for any theoretical analysis. Given the lack of access to model internals or detailed pretraining corpora, we intentionally refrain from speculative causal claims.
>
>
>
> **Finance Domain Models**
>
> We appreciate this suggestion. As noted in Appendix F (lines 626–633), we did include FinMA-7B—a finance-specialized LLM—in our evaluation. However, the model frequently failed to produce usable outputs (e.g., returning empty strings or numbers without units), limiting its reliability for large-scale analysis. That said, we would be happy to move these results into the main paper in the camera-ready version if the reviewer believes it would improve clarity and completeness.
>
> We also note in our Ethics section that many prominent finance-specific LLMs (e.g., BloombergGPT) are not publicly available for inference, making them difficult to evaluate at scale. We hope our framework will be a useful tool for assessing such models as they become more accessible.
>
>
>
> We hope that the breadth of our empirical analysis, combined with the clarity of our framing makes a compelling case for acceptance.

---

> > ### Author Response · Authors · 2025-06-07
> >
> > We hope our previous response addressed your questions and concerns satisfactorily. Please let us know if there is anything further we can clarify. We would greatly value your feedback.

---

> > > ### Comment · Reviewer_2yqz · 2025-06-09
> > >
> > > thanks for the clarifications.
> > > I don't have further comments.

---

> > > > ### Author Response · Authors · 2025-06-09
> > > >
> > > > Thank you for your engagement. We just wanted to reiterate for completeness that we did include a finance-specific LLM (FinMA-7B) in our study, as discussed in Appendix F (lines 626 to 633). While the model's outputs were often incomplete, we would be happy to surface these results in the main paper if helpful. We hope the overall contributions of this work, including our large-scale dataset and stratified bias analysis, are clear and meet the bar for acceptance.

---

### Official Review · Reviewer_hcdm · 2025-05-12

**Rating:** 8
**Confidence:** 4
**Ethics Flag:** 1

**Summary:**

The paper makes a valuable contribution by evaluating the financial knowledge of prominent Large Language Models (LLMs) and focusing on two kinds of bias: temporal (knowledge gaps variation across time) and cross-sectional (variation across firms sizes). Using a new dataset of over 197,000 revenue-related prompts covering 17,000+ U.S. public companies from 1980 to 2022, the authors benchmark several frontier LLMs, including GPT-4, Gemini, DeepSeek-V3, and Llama-3. The findings reveal systematic weaknesses in LLMs: they perform better on larger companies and more recent data, but paradoxically also hallucinate more on those same entities. This is one of the few papers I am aware of evaluating factuality and correctness of LLMs in narrow domains.

**Questions To Authors:**

See the Reasons to Reject section.

**Reasons To Accept:**

- Novel dataset:
  - the paper introduces a semi automatically generated Revenue Prompt Dataset, spanning multiple decades and company profiles,
    enabling a high quality evaluation.
  - the inclusion of auxiliary firm-level features (market capitalization, retail/institutional attention, readability) enhances the analysis.
- Rigorous evaluation:
  - the ternary outcome classification (correct, hallucination, no answer) is interesting and quantitatively robust.
  - logistic regression analysis helps isolate key influencing factors on LLM behavior.
- Clear presentation and high quality argument:
  - demonstration of a retrograde knowledge bias adds a new dimension to the discussion of LLM limitations.
  - the identification of a paradoxical phenomenon - higher accuracy and hallucination likelihood in high-market-cap companies - adds to
    the discussion of important questions about model confidence and reliability.
- Potential broader relevance:
  - the evaluation framework is applied outside finance illustrating potential for broader domain adaptation.

**Reasons To Reject:**

- Limited evaluation for full dataset:
  - high-performing models like GPT-4.5, Gemini, etc. are only evaluated on a subset of the data due to API cost constraints. This leads me
    to wonder about the generalizability of those results, albeit I of course understand the reasoning.
- Narrow focus:
  - the analysis is limited only to revenue-based questions. This does not capture how LLMs perform on more complex financial reasoning
    tasks (i.e. interpreting cash flows, earnings reports, etc.) I would like to have seen one or two other, qualitatively different questions to
    make the analysis more comprehensive.
- Lack of causal explanation:
  - while the retrograde bias is well-documented, causal analysis (ablation studies, retraining experiments) on why these biases exist is
    absent. I am guessing part of the reason for this behavior is the change in volume of training data over time (the amount of machine
    readable text has increased drastically since the 90s, not to mention the 80s). I don't have an explanation for the paradoxical behavior
    where rate of hallucinations goes up with time. Some hypotheses on this would be good to see.

---

> ### Author Response · Authors · 2025-05-31
>
> We sincerely thank the reviewer for their thoughtful and constructive feedback. We are especially grateful for the recognition of our work’s contributions—both in terms of dataset novelty and the analysis of LLM behavior along temporal and cross-sectional dimensions. We also appreciate the reviewer’s insightful observations about the paradoxical trends we document, as well as the broader applicability of our evaluation framework. In what follows, we address the concerns raised, particularly around full dataset usage, scope, and the need for more discussion on the causal underpinnings of observed biases.
>
>
>
> **Limited evaluation for full dataset**
>
> Thank you for your understanding regarding the API cost constraints. As noted, we evaluated expensive models like GPT-4.5 and Gemini on a representative subset, while running full-sample evaluations for open or lower-cost models like Llama-8B, Llama-70B, GPT-4o-mini and GPT-4o. Encouragingly, the key trends—such as retrograde bias and cross-sectional disparities—are consistent across both subsets and full-sample evaluations, supporting the robustness of our findings.
>
>
>
> **Narrow focus**:
>
> We appreciate the suggestion to broaden the scope beyond revenue-based questions. While our core analysis focuses on revenue to enable clean evaluation of factual knowledge, we emphasize in Section 5 and Appendix L that our framework is generalizable across domains—we illustrate this by applying it to Spanish soccer league statistics. Additionally, in Appendix K, we explore multi-turn prompting for stock recommendations, showing how revenue knowledge gaps influence downstream financial decisions. These extensions underscore the broader relevance and applicability of our evaluation approach.
>
> **Causal explanation**:
>
> We agree that understanding the root causes of retrograde and paradoxical hallucination biases is an important direction. However, with the lack of access to pre-training data—even for open-weight models—causal analysis (e.g., ablations or retraining) is not feasible. That said, we hypothesize these trends stem from disparities in pre-training data availability: recent years and prominent companies are more widely discussed and thus likely overrepresented. We partially support this through correlations with SEC access logs and retail/institutional attention, but refrain from making causal claims without further evidence.
>
>
>
> We hope these clarifications highlight the broader relevance and rigor of our work, and we thank the reviewer again for the thoughtful engagement.

---

> > ### Author Response · Authors · 2025-06-07
> >
> > We sincerely appreciate your thoughtful and encouraging review. We hope our responses further clarify the contributions and scope of the work. We’d value any additional feedback you may have.

---

> > > ### Comment · Reviewer_hcdm · 2025-06-09
> > >
> > > The authors' comments address my questions.

---

> > > > ### Author Response · Authors · 2025-06-09
> > > >
> > > > Thank you, we’re glad to have addressed your concerns!

---

### Official Review · Reviewer_YwGF · 2025-05-13

**Rating:** 7
**Confidence:** 4
**Ethics Flag:** 1

**Summary:**

The paper is exploring an interesting direction for the use of LLMs in the market/stock data understanding in terms of understanding the depth (time wise) of the financial data embedded into the training of the LLM. The authors probe state-of-the-art models (GPT-4o,4.5, Gemini 1.5 Pro, DeepSeek-V3, Llama-3 small and large) with 197k questions related to the financial data from various benchmark datasets (all in a zero shot setting). The authors explore the knowledge gap in the models (in terms of timing, old vs new), biases in terms of favoring large companies over smaller ones, and any hallucinations that may occur given the research is into numbers.

**Questions To Authors:**

1. What is the reason for the drip in 2021-2022 (line \#200), is it due to COVID-19? But even then, COVID-19 was peaked in 2020\.

**Reasons To Accept:**

1. The authors explore a challenging research domain for LLM given the factual information given in the task with the financial datasets, they also study the impact of the LLM hallucinations in the predictions.
2. In terms of evaluation, the work is evaluated across a large number of datasets covering 197k instances but across a diverse range of financial related dataset sources.
3. The authors have utilized an evaluation setup (Figure 2\) based on zero short prompting, that not only evaluates/probes the LLM to be factual but also to be accurate in terms of time.
4. The authors have also sub-sectioned their findings section to make it an easy parse for the readers.
5. The supplemental data section is also rich in terms of including all the information needed to reproduce their experiments with the prompt templates and additional results in terms of empirical results into their findings.

**Reasons To Reject:**

1. A strength of the paper is also a weakness given the limited scope of narrowing down the market data to only US related data since some of the companies are international and financial data is impacted by international events.
2. The authors have explored only zero-shot prompting, this is the challenge for LLMs but it would have been helpful to see how it does change if there was one-shot prompting.
3. Seems like not all models are evaluated with all the 197k instances and the large closed sourced models are limited to a sub-sampled dataset due to cost constraints. The highlight of the paper should have been to this smaller dataset in such case.

---

> ### Author Response · Authors · 2025-05-31
>
> We appreciate your thoughtful and constructive review. Your recognition of our exploration into the depth and bias of financial knowledge within LLMs—especially under a zero-shot setup across a large and diverse set of financial QA instances—is encouraging. We are particularly grateful for your acknowledgment of our systematic evaluation pipeline, extensive dataset design, and reproducibility practices. Below, we address your concerns regarding scope, prompting strategies, dataset coverage across models, and your question about the 2021–2022 dip.
>
> **Geographic Scope**:
>
> Thank you for pointing this out. Our study focuses on *U.S. publicly traded* companies, which includes many international firms such as Toyota from Japan (TM), Alibaba from China (BABA), Infosys from India (INFY) and many more. Additionally, many top US companies have most of their revenue coming from outside the US and are heavily influenced by global events. Moreover, as discussed in Section 5 and Appendix L, our evaluation framework is generalizable beyond finance—we also demonstrate its applicability in assessing knowledge biases in domains like Spanish soccer.
>
>
>
> **Prompting Strategies**:
>
> We use zero-shot prompting to avoid injecting answer-related bias (e.g., via revenue in one-shot examples). Manual checks (Appendix D) confirm that models understood the prompt well. Additionally, we explore more sophisticated prompting strategies in Appendix K.
>
>
>
> **Dataset Coverage Across Models**:
>
> Due to API cost constraints (e.g., $3,000 for full-sample GPT-4.5), we ran expensive models on a *representative subset* ($120). In contrast, we evaluated open or lower-cost models Llama-8B, Llama-70B, GPT-4o-mini and GPT-4o on the full dataset. Importantly, our key findings remain consistent across models.
>
>
>
> **Question about the 2021–2022 Dip**:
>
> We discuss this anomaly in lines 199–204 and agree it merits further study. Emerging work [1] explores why LLMs may misalign with their stated cutoffs. Our study complements this by going beyond cutoffs—probing retrograde and cross-sectional knowledge gaps. We view this as a growing area for future research.
>
>
> [1] Cheng, Jeffrey, et al. "Dated data: Tracing knowledge cutoffs in large language models." arXiv preprint arXiv:2403.12958.
>
>
> We appreciate your thoughtful engagement with our work. We hope our clarifications address your concerns and reinforce the broader significance and rigor of our study—both in methodology and in its implications across domains.

---

> > ### Author Response · Authors · 2025-06-07
> >
> > We hope our previous response addressed your questions and concerns satisfactorily. Please let us know if there is anything further we can clarify. We would greatly value your feedback.

---

### Decision · Program_Chairs · 2025-07-08

**Decision:**

Accept

**Comment:**

This paper presents a large-scale empirical study evaluating the financial knowledge of several state-of-the-art Large Language Models (LLMs). The authors have constructed a dataset of over 197,000 revenue-related questions spanning four decades and over 17,000 U.S. public companies. The core contribution lies in systematically probing for and identifying two key biases: a "retrograde" temporal bias, where models perform better on more recent data, and a cross-sectional bias, where models are more accurate for larger, more prominent companies.

Consolidated Strengths
* Novel and Valuable Dataset: All reviewers acknowledged the significant effort and value in creating the 197k-instance Revenue Prompt Dataset. This is seen as a durable contribution to the community, enabling future research on LLM factuality in specialized domains.
* Rigorous and Large-Scale Evaluation: The scale of the experiment is a clear strength. The systematic approach, clear evaluation setup (including the ternary outcome classification of correct/hallucination/no answer), and subsequent regression analysis were praised for their robustness.

Consolidated Weaknesses
* Limited Scope: A key criticism centered on the study's scope. The focus is exclusively on U.S. companies, revenue-based questions, and a zero-shot prompting strategy.
* Generalizability and Model Choices: The fact that the most advanced closed-source models were only tested on a subset of the data due to cost constraints was a recurring concern. Additionally, Reviewer 2yqz pointed out that the study could have been strengthened by including finance-specialized LLMs, not just generalist ones.

Conclusion
This paper makes a valuable empirical contribution to a critical and underexplored area: the factuality of LLMs in the financial domain. The creation of a large-scale, focused dataset is a good contribution itself. The findings on temporal and size-based biases are clear, well-supported, and add an important, nuanced discovery about the nature of hallucinations in the financial area. The majority of reviewers rightfully identify it as an accept.